# Habilitation of Executive Functions in Pediatric Congenital Heart Disease Patients through LEGO^®^-Based Therapy: A Quasi-Experimental Study

**DOI:** 10.3390/healthcare10122348

**Published:** 2022-11-23

**Authors:** Eduardo Espinosa-Garamendi, Norma Angélica Labra-Ruiz, Lizbeth Naranjo, Claudia Andrea Chávez-Mejía, Erika Valenzuela-Alarcón, Julieta Griselda Mendoza-Torreblanca

**Affiliations:** 1Unidad de Habilitación Cognitiva, Subdirección Médica, Instituto Nacional de Pediatría, Ciudad de México 04530, Mexico; 2Laboratorio de Neurociencias, Subdirección de Medicina Experimental, Instituto Nacional de Pediatría, Ciudad de México 04530, Mexico; 3Departamento de Matemáticas, Facultad de Ciencias, Universidad Nacional Autónoma de México, Ciudad de México 04510, Mexico; 4Escuela de Psicología, Universidad Panamericana, Ciudad de México 03920, Mexico; 5Fundación Care & Share for Education, Ciudad de México 01050, Mexico

**Keywords:** executive functions, LEGO^®^-based therapy, congenital heart disease, orbitomedial cortex, RACHS1 scale, cognitive habilitation

## Abstract

Congenital heart disease is defined as an abnormality in the cardiocirculatory structure or function. Various studies have shown that patients with this condition may present cognitive deficits. To compensate for this, several therapeutic strategies have been developed, among them, the LEGO^®^ Education sets, which use the pedagogic enginery to modify cognitive function by didactic material based on mechanics and robotics principles. Accordingly, the goal of this study was to evaluate the effect of cognitive habilitation by using LEGO^®^-based therapy in pediatric congenital heart disease patients. This was a quasi-experimental study; eligible patients were identified, and their general data were obtained. In the treatment group, an initial evaluation with the neuropsychological BANFE-2 test was applied; then, once a week, the interventions were performed, with a final test at the end of the interventions. In the control group, after the initial evaluation, a second appointment was scheduled for the final evaluation. Our results show that >50% of children presented cognitive impairment; nevertheless, there was an overall improvement in treatment patients, showing a significant increase in BANFE scores in areas related to executive functions. LEGO^®^-based therapy may be useful to improve cognitive abilities; however, future research should be performed to strengthen the data.

## 1. Introduction

Congenital heart disease is defined as an abnormality in the cardiocirculatory structure or in the function that includes the heart and the great vessels; it occurs during embryonic and fetal development and is present at birth, although it is discovered later during the life of the carrier of the malformation [1]. Worldwide, it is considered one of the most frequent congenital anomalies with the highest mortality, since it is estimated that 9 out of 1000 live newborns are affected. Thus, 1.35 million babies are born each year with some type of heart disease, which causes more deaths in the first year of life than any other birth defect [2]. In the United States, it affects 36,000 infants per year [3], and in Mexico, there are between 18,000 and 20,000 new cases per year [4].

Various studies have shown that patients with congenital heart disease may present cognitive deficits [5,6,7] that are related both to hypoperfusion caused by heart malformation, which causes low oxygenation performance in the frontal and prefrontal cortex [8,9], as well as the factors involved during cardiovascular surgery, one of the main procedures to treat this disease [10].

Cardiovascular surgeries can be stratified through the Risk Adjusted Classification for Congenital Heart Surgery (RACHS-1), which considers the complexity level at the congenital malformation and the surgery risk, among other factors [11]. It has been reported that a higher level on the RACHS-1 scale (from 3 to 6) may result in a higher detriment at a cognitive level in these patients [12,13].

In children with congenital heart disease, the prefrontal region of the brain is affected, which could be related cognitively to executive functions, which are a series of capabilities to control, elaborate goals, planning, do and regulate behavior and cognitive processes effectively [14,15]. By showing affectations in executive functions, such as working memory and the response inhibition process [16], different learning sequences may be altered, including reading-writing functions and mathematical progression. Depending on the stage of life, it could increase the probability of manifesting a higher deterioration in cognitive functions and, consequently, in emotional, behavioral and social functions, limiting his/her quality of life [17].

To compensate for cognitive deficits, therapeutic strategies centered on cognitive habilitation have been developed; their main objective is the growth in cortical plasticity by functional retraining, stimulating the cognitive system to improve or activate this process [18,19]. LEGO^®^ Education are part of these strategies; they use a pedagogic enginery to modify the cognitive function sequence through didactic material based on mechanics and robotics principles [20,21].

Diverse studies have reported an improvement in social and communication skills, specific conduct and family relationships in children and younglings with an autistic spectrum who were treated with LEGO^®^-based therapy [21,22]. Likewise, children from 4 to 10 years, diagnosed with cerebral paralysis and related motor affections, showed interest in the LEGO^®^ robots, showing changes in their behavior and in their social and language abilities after the therapy [23]. Additionally, Lindsay and Lam (2018) [24] reported that children with different kinds of disabilities showed progress from solitaire plays to parallel and/or cooperative plays as the LEGO^®^ robotic program progressed, helping the children’s capacity to interact with their mates and facilitating their social development. However, few studies have reported the use of this tool to improve the cognitive deficit of congenital heart disease patients [25]; thus, the goal of this investigation was to measure the effect of cognitive habilitation by the intervention of assembly and robotic programming with LEGO^®^-based therapy on the executive functions of pediatric congenital heart disease patients.

## 2. Methodology

Quasi-experimental study in children with a diagnosis of congenital heart disease undergoing heart surgery. The sample selection was at convenience, intentional and with voluntary participation. The protocol was approved by the Research and Ethics Committee of the National Pediatrics Institute (INP; registration number 2020/51).

### 2.1. Participants

Forty-five patients diagnosed with congenital heart disease who underwent cardiovascular surgery were recruited. Of these 45 patients, 24 remained in the study; 14 were part of the experimental group, and 10 were part of the control group. The children’s ages ranged from 6 to 17 years. Children suffering from any relevant psychiatric or psychological disorder were excluded. Patients (or their parents) were free to withdraw from the study at any time.

### 2.2. Instruments

#### 2.2.1. RACHS-1

The complexity of the surgical intervention was categorized using the RACHS-1 scale. This method allowed the evaluation of the surgical risk depending on the type of heart disease, the type of repair and some other elements that may influence the final result, such as weight, age and associated abnormalities; it is divided into 6 levels or risk categories: level 1 is the least complex, and level 6 is the most [11,26].

#### 2.2.2. BANFE-2

To measure executive functions, the Battery of Executive Functions (BANFE-2), in its original version, was applied. This instrument was used for the evaluation of cognitive processes that depend mainly on the orbitomedial cortex, the anterior prefrontal cortex and the dorsolateral cortex. The orbitomedial cortex assesses motor control, inhibitory control and risk selection; the anterior prefrontal cortex is associated with abstract meaning, metamemory and metacognitive control; and the dorsolateral cortex assesses working memory, visuospatial working memory, consecutive and inverse operations, planning, visuospatial planning, cognitive flexibility, productivity of abstract thinking, verbal fluency and sequential planning [27].

#### 2.2.3. LEGO^®^ Education Scale

The execution of the intervention was measured with the LEGO^®^ scale, a Likert-type scale, developed by the researchers according to the type of behavioral execution. It was based on executive functions [28] and was classified as follows: 0 = does not execute; 1 = difficult to execute; 2 = executes and 3 = executes with ease. To quantify the validity coefficient of the scale, the Hernández-Nieto (2002) procedure [29] was used, with qualifying scores of 0.80 to 0.90, indicating an acceptable validity and concordance coefficient.

### 2.3. Process

Eligible patients were identified and invited to participate. Parents or guardians and patients were provided with information about the study and were asked to sign the informed consent letter. For their registration, the relevant clinical and demographic data were considered, and the complexity of their surgical intervention was classified according to the RACHS-1 scale. Subsequently, a startup session was held to assess and determine the degree of cognitive deficit using the BANFE-2 neuropsychological test (lasting 60 min). In the treatment group, in the second stage, the children were scheduled once a week to carry out between seven and eighteen 60-min interventions (Table 1; Appendix A), assessing their execution using the LEGO^®^ Education scale. At the end of all the interventions, a postintervention evaluation was applied using the BANFE-2 test. In the control group, after the initial evaluation, a second appointment was scheduled for the final evaluation. Assessments and interventions were carried out by neuropsychologists trained in LEGO^®^ Education.

### 2.4. Intervention Evaluation

The intervention was designed based on the conceptualization and operationalization of neuropsychological variables of the BANFE-2 [27,28], the LEGO^®^-based therapy [30] and the basic frontal function enablement model [25]. The sessions were carried out individually in the Cognitive Habilitation Unit within the INP facilities. In the 60-min interventions, different functions were worked on in the same session, increasing the complexity of the exercises each week. The children took between seven and eighteen sessions depending on their ability to assemble the more complex robots, i.e., those children who assembled the robots faster took fewer sessions (Table 1; Appendix A).

### 2.5. Statistical Analysis

The scores for patients in the experimental setting were analyzed using the loess smoothing method [31] to estimate the mean of profiles for the scores by RACHS-1. The dependent variables studied were orbitomedial cortex, anterior prefrontal cortex, dorsolateral cortex, and total executive functions. The hypothesis scored with significant differences between the control and treatment groups was examined using analysis of covariance (ANCOVA) [32,33] followed by a Bonferroni test and the Wilcoxon rank sum test [34]. Statistical analysis was performed using R version 3.4.1 and R Studio version 0.99.902 software. We considered a *p* value < 0.05 to be statistically significant.

## 3. Results

### 3.1. Descriptive Analysis of the Population

Table 2 shows a summary of the descriptive characteristics of the study population. In total, we analyzed 24 patients, 10 in the control group (six male and four female) and 14 in the treatment group (eight male and six female). Their average age was 9.5 for the control group (interquartile range, IQR 8.0–12.25) and 8.0 for the treatment group (IQR 6.25–10.0). The stratification with the RACHS-1 scale was as follows: in the control group, five patients were classified as level one; three patients as level two; one patient as level three; and one patient as level four. In the treatment group, one patient was classified as level one; two patients as level two; five patients as level three; and six patients as level four.

### 3.2. Cognitive Conditions of Pediatric Patients with Congenital Heart Disease

The cognitive condition of pediatric patients with congenital heart disease before and after LEGO^®^-based therapy is shown in Table 3. The evaluations were focused on the orbitomedial, anterior prefrontal and dorsolateral cortices and on total executive functions. According to the battery of executive functions, the scores increased in nine patients, giving clinical change in diagnoses, whereas in two patients, the score increased, but not the diagnosis. In the orbitomedial cortex, an increase in the score and diagnostic range change were observed in 10 patients, with an increase in the score in one subject, without diagnostic change. In the anterior prefrontal cortex, there was an increased score and diagnostic change in seven participants. Finally, in the dorsolateral cortex, there was an increased score, and the diagnostic range changed in five patients.

Likewise, the cognitive condition of control pediatric patients with congenital heart disease is shown in Table 4. Notably, in the total executive functions, only one patient changed the diagnostic range, and three patients increased their score very slightly. In the orbitomedial cortex, two patients had slightly increased scores. In the anterior prefrontal cortex, an improvement in diagnosis was observed in five patients, and two children increased their scores. Finally, in the dorsolateral cortex, a change in diagnostic range was observed in one subject, and a very slight score change was observed in five patients.

### 3.3. Evaluation of LEGO^®^-Based Therapy in Pediatric Patients with Congenital Heart Disease

The children’s execution in LEGO^®^-based therapy was evaluated with the LEGO^®^ scale, also considering their classification on the RACH-1 scale. Figure 1 shows the execution percentages by child. The plot on the top shows the profiles of the execution. Each line represents a patient and is colored by his/her RACHS-1 classification. The plot on the bottom shows the mean of profiles (solid line) computed with the loess smoothing method [31] and their 95% confidence intervals (shaded areas), and the colored line represents the RACHS-1 classification.

In general, the observed scores are higher as sessions progress. The profiles for patients with RACHS-1 in classifications one or two have higher scores than those with classifications three or four. Moreover, in the first sessions (sessions 1 to 5), the scores of the patients show significant differences between patients with RACHS-1 in levels one or two and patients with RACHS-1 in levels three or four, since the 95% confidence intervals for these sessions do not intersect (Figure 1 bottom). However, as the sessions progress, these differences are not so considerable. For sessions 6 to 10, there were no significant differences between patients with RACHS-1 levels one or two and those with RACHS-1 level three, since their 95% confidence intervals intersect; the only significant difference was shown between patients with RACHS-1 levels one or two and those with RACHS-1 level four since their 95% confidence intervals did not overlap (Figure 1 bottom). Finally, from sessions 11 to 14, there were no significant differences between patients according to RACHS-1, although patients with RACHS-1 levels three or four had average scores lower than those with RACHS-1 levels one or two; the advance in LEGO^®^-based therapy execution seemed closer as sessions progressed (Figure 1 bottom).

### 3.4. Overall Improvement of Pediatric Patients with Congenital Heart Disease with LEGO^®^-Based Therapy

To evaluate the overall improvement of congenital heart disease patients with LEGO^®^-based therapy, we used 2 × 2 contingency tables constructed with categorized variables from the BANFE-2 pretest and posttest normalized scores of each group. Two categories were established. If the total scores were greater than 84, they were categorized as normal; otherwise, they were categorized as alteration. Figure 2 summarizes the variables related to executive functions of the orbitomedial cortex, anterior prefrontal cortex and dorsolateral cortex and the total executive functions for each group (control or LEGO^®^-based therapy).

It was observed that, in all cases, the patients who had a normal category in the pretest remained in the same category in the posttest. However, in the control group, most of the patients who showed an alteration in the pretest continued with an alteration in the posttest (with the exception of the anterior prefrontal cortex). Furthermore, most patients who received LEGO^®^-based therapy changed from alteration to normal category (with the exception of the anterior prefrontal cortex); for instance, in the orbitomedial cortex, in the control group, the four subjects categorized as alteration in the pretest continued in alteration in the posttest; however, for the treatment group, two subjects categorized as alteration in the pretest continued in alteration, but four subjects categorized in alteration improved to normal category.

### 3.5. Effect of LEGO^®^-Based Therapy in Pediatric Patients with Congenital Heart Disease

Figure 3 shows the summary of the BANFE-2 scores for the control and LEGO^®^-based therapy patients. Plots on the left show the boxplot of the normalized BANFE-2 scores by test and group. Plots on the right show the estimated marginal means of the posttest scores by group, under the mean of the pretest, showing an increase in the LEGO^®^-based therapy group in all evaluated areas, with the exception of the anterior prefrontal cortex. The significant BANFE-2 scores increased in the treatment group after LEGO^®^-based therapy in the orbitomedial cortex and in the total executive functions.

### 3.6. Analysis of Gain Score

Another procedure to analyze a pretest-posttest design compares the gain scores [32], which evaluates the differences between pretest and posttest for each variable, i.e., gain score = posttest score − pretest score.

Figure 4 shows the boxplots for the gain scores for the evaluated variables by group. For the anterior prefrontal cortex, the boxplots look similar, which means that there is no difference in the gain score between the control and LEGO^®^-based therapy groups. Something similar happens for the dorsolateral cortex; boxplots do not show differences between the control and treatment groups; however, there is a slight increase (improvement) in the latter.

Remarkably, for the orbitomedial cortex, the boxplots of the control and LEGO^®^-based therapy groups were significantly different; for the control, the median was 0, and the first and third quartiles were −3 and 0, but for the treatment, the median was 31.5, and the first and third quartiles were 11 and 44, respectively, which means that for the orbitomedial cortex, children improved their scores on the BANFE-2 neuropsychological test and, therefore, their cognitive abilities. Something similar happens for total executive functions; for control, the median is 0.5, and the first and third quartiles are zero and two, respectively, while for treatment, the median is 26, and the first and third quartiles are 5 and 61, respectively. Then, the boxplot shows a significant difference between control and treatment, with the posttest scores in treatment being higher than those in pretest.

## 4. Discussion

In this research, we evaluated the cognitive deficit presented by patients with congenital heart disease after undergoing cardiovascular surgery and the effect of therapy based on LEGO^®^ Education on frontal executive functions. In the pretest measurement, more than 50% of the participants showed cognitive impairment, either severe or mild to moderate, which corresponds to what is stated in the literature regarding cognitive impairment in patients with congenital heart disease [12,13].

Additionally, we observed that the degree of complexity of the pathology influenced the execution performance of the therapy based on LEGO^®^ Education, since the patients with classification one or two of RACHS-1 obtained higher scores in the execution and required fewer sessions to achieve better performance. In comparison, heart disease patients with RACHS-1 classification three or four had more difficulty in execution and required greater cognitive effort, which involved a greater number of sessions. Thus, the patients most affected by the hypoperfusion sequence and associated surgical factors presented important deficiencies that led to psychological implications [16]. The above could even have repercussions in the academic area due to alterations in learning sequences, which can hinder mathematical and reading-writing skills [35].

Although the affected areas in patients with heart disease occur mainly in the frontal and prefrontal regions, the results showed that it is possible to generate significant changes in the orbitomedial cortex. This indicates that the patients reached a higher level of development than expected for their age in the detection of risk selection, working memory, motor control and inhibitory control [15], implying that basic frontal functions improved from the diagnostic range after the intervention, so it would be expected that this cortex and its functions are working in an optimal process. On the other hand, it was observed that the control group maintained the same diagnosis or increased the coded score slightly, which could mean that the neurons are in their natural maturation process for the age range. Therefore, therapy based on LEGO^®^ Education can favor the group of neurons and processes associated with this cortex to develop more quickly, in accordance with the concepts of concentration, mutual induction and cell irradiation proposed by physiology behavior and the principles of cortical plasticity [36].

Regarding the functions related to the dorsolateral cortex, the scores were not significant, but there was a tendency to have an improvement after the treatment in most of the patients in the therapy based on the LEGO^®^ Education group, unlike the control patients, whose score increased only slightly. This means that the experimental group had a tendency to increase their capacity in working memory, planning, cognitive flexibility, verbal fluency and abstract thought productivity in relation to the control group [15,25]; however, scores could vary due to age ranges and different cardiovascular diagnoses [27,37]. It would be important, in a second approach, to be able to count on a greater number of patients.

Following the development of executive functions, the ranges of improvement in the diagnosis, obtained at the end of the interventions, in the functions related to abstract meaning, monitoring and metacognitive control, which correspond to the anterior prefrontal cortex [15,28], suggest that the patients could maintain proper functioning in the anterior prefrontal cortex and its functions for adolescence and adulthood, which is when the development of this structure culminates.

In relation to other studies, the ascending curve of the execution scale shows that the stimulation of neuronal groups, in this case associated with executive functions with LEGO^®^-based therapy, promotes their cognitive habilitation process, which coincides with the results of [38,39], which report strategies related to the assembly, programming and elaboration of abstract challenges related to computerized or noncomputerized therapies; however, therapy based on LEGO^®^ Education for executive functions involves multiple cognitive and behavioral processes, as well as computerized and noncomputerized activities, merging these processes in the same intervention [40]. In addition, in previous work, we reported that cognitive intervention, with therapy based on LEGO^®^ Education for executive functions in congenital heart disease patients, had a favorable impact on enabling basic frontal functions, showing changes mainly in working memory and visuospatial and verbal fluency, which are also related to the orbitomedial and dorsolateral cortices [25].

It is worth mentioning that most of the patients started with low execution in the first sessions, and as they progressed, they obtained greater control of the material, with greater motor capacity and in less execution time, thus increasing their scores. Additionally, as the patients progressed through the sessions, they were able to solve more complex programming and robotics problems. In the two cases in which there was no improvement, this could be because the patients finished the program, but sometimes, they missed some interventions, and with this, we observed that they could not maintain working memory or inhibitory control as the patients who were constant in all their sessions. The constancy of work helps the neuronal group associated with executive functions to remain constantly stimulated, achieving greater concentration and irradiation, resulting in a long-term effect.

Additionally, it is important to point out the limitation of this study due to the number of patients. Since this work was carried out shortly before and during the COVID-19 pandemic, many patients who started therapy were unable to continue due to various factors intrinsic to the situation. However, our results showed interesting trends and significant differences that can be corroborated and strengthened in future studies.

Finally, children’s parents reported that participants improved their manual motor skills and interaction behaviors, for example, interpersonal relationships. The above could be related to the improvement of skills in the social sphere, as reported in the population with autism and Asperger’s [22,41,42,43,44,45,46,47,48]. Therefore, for future studies, it will be important to include socioemotional factors, emotional self-control and motor skill scales in the evaluation.

## 5. Conclusions

Despite the limited number of patients in this study, we observed that cognitive habilitation through LEGO^®^-based therapy in congenital heart disease patients might generate significant changes in areas related to executive functions, which are impaired due to the disease itself and its treatment. By increasing the functions associated with the frontal and prefrontal cortex at school age, children could improve their cognitive abilities, which may favor learning in academic and functional areas of daily life. In addition, since this was a quasi-experimental study, future research should be done to strengthen and corroborate the data.

## Figures and Tables

**Figure 1 healthcare-10-02348-f001:**
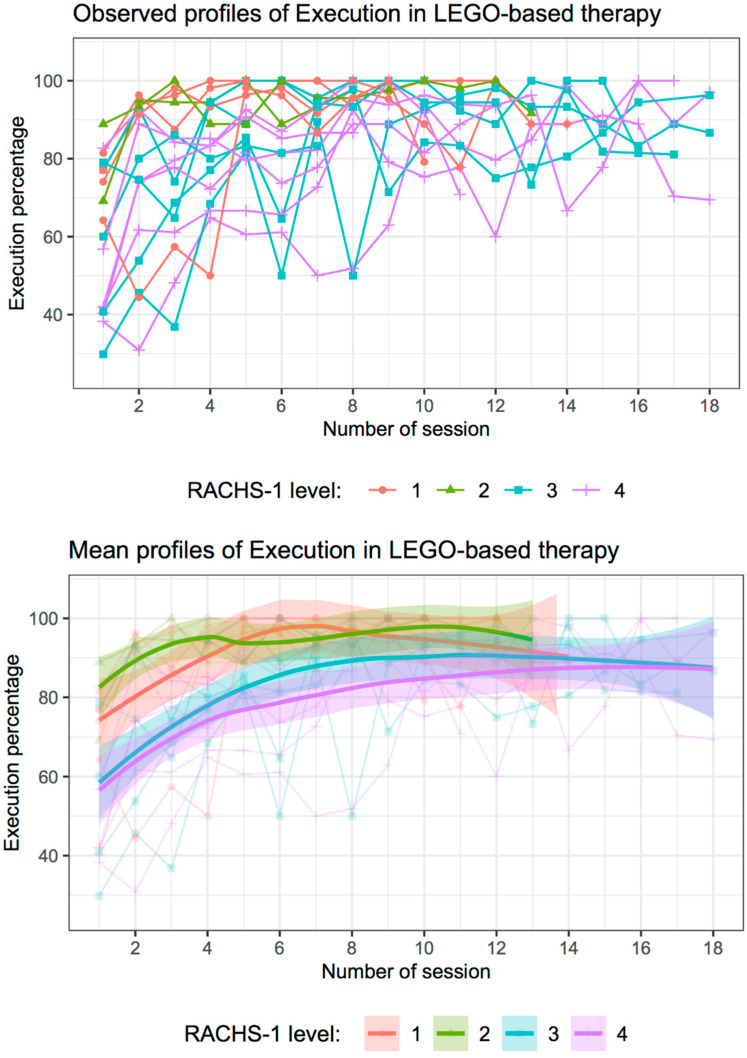
Evaluation of execution of pediatric patients with congenital heart disease in LEGO^®^-based therapy by sessions and RACHS-1 classification. (**Top**): Observed profiles of each patient colored by his/her RACHS-1 level. (**Bottom**): Mean of profiles computed with the loess smoothing method (solid line) and 95% confidence intervals (shaded areas); colored lines represent the RACHS-1 level.

**Figure 2 healthcare-10-02348-f002:**
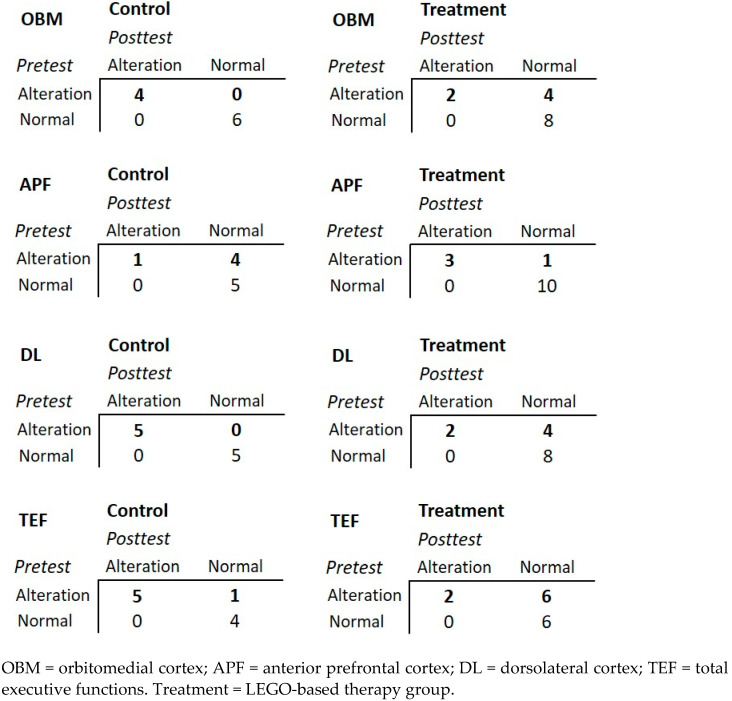
Overall improvement of congenital heart disease pediatric patients.

**Figure 3 healthcare-10-02348-f003:**
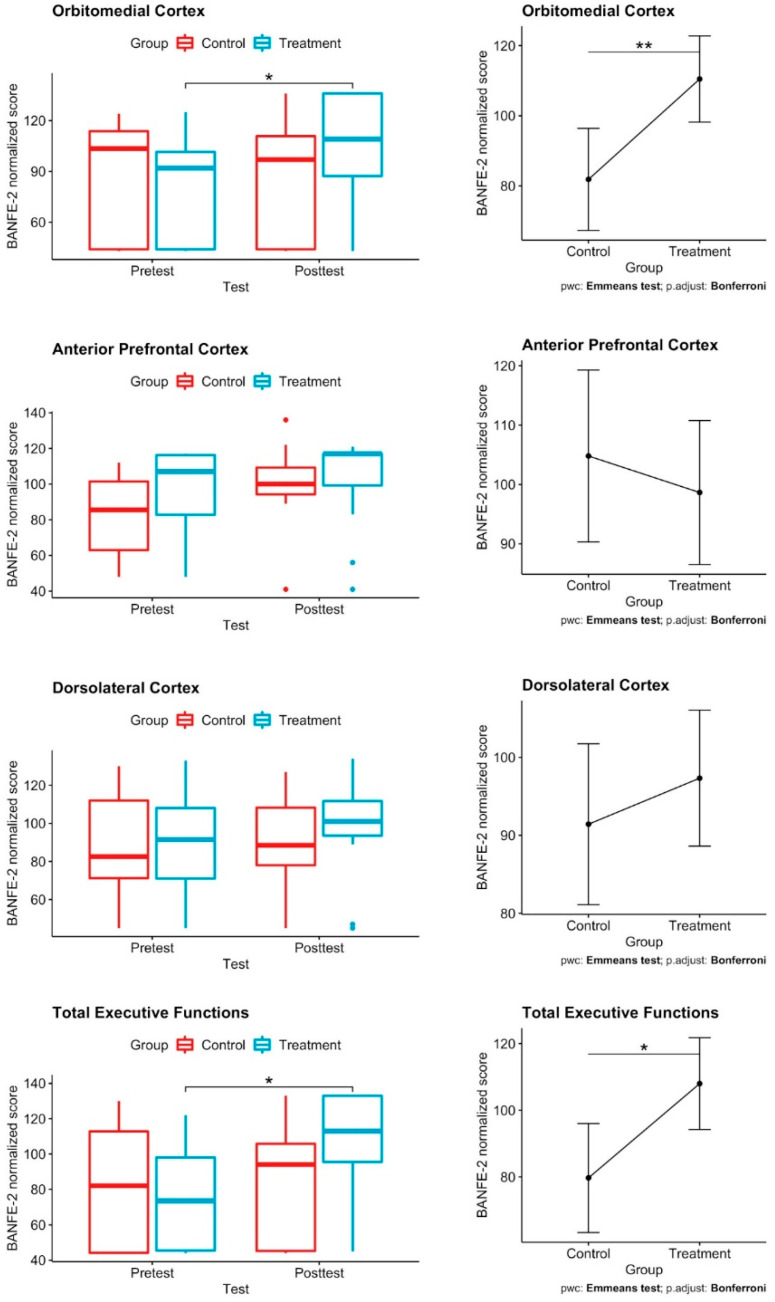
Effect of LEGO^®^-based therapy in pediatric patients with congenital heart disease. **Left**: Box plots depict changes in BANFE-2 scores of congenital heart disease pediatric patients in the control and treatment groups; there was a significant score increase after LEGO^®^-based therapy in the orbitomedial cortex and in the total executive functions. ANCOVA test; * *p* ≤ 0.050, ** *p* < 0.01. **Right**: ANCOVA followed by Bonferroni post hoc test applying multiple testing correction; there is an increase in the adjusted mean score in the LEGO^®^-based therapy group in all evaluated areas, with the exception of the anterior prefrontal cortex. Control *n* = 10; treatment *n* = 14.

**Figure 4 healthcare-10-02348-f004:**
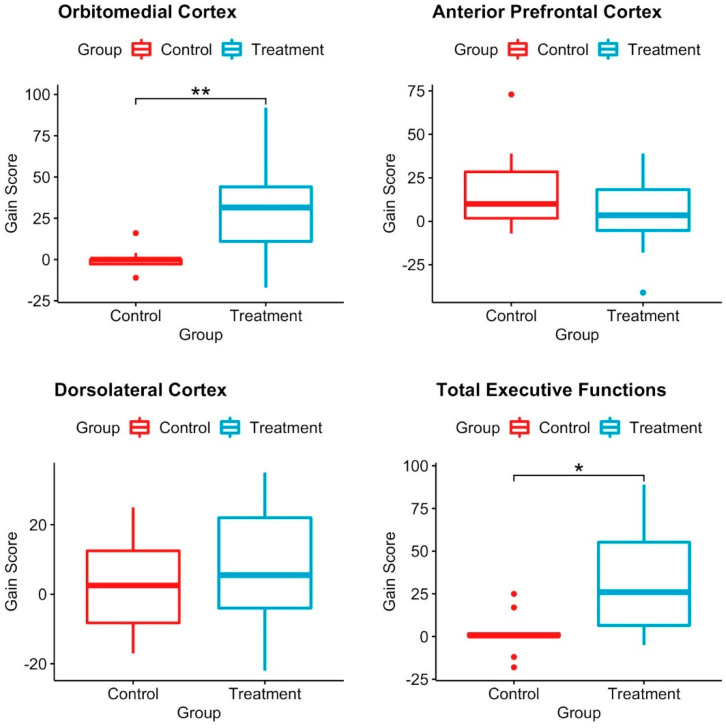
Boxplot of the gain scores for the diagnosis variable by group. Box plots depict the differences between pretest and posttest gain scores in the control and LEGO^®^-based therapy groups. There were significant differences between groups in the orbitomedial cortex and in the total executive functions. Wilcoxon test; * *p* ≤ 0.01; ** *p* ≤ 0.001. Control *n* = 10; treatment *n* = 14.

**Table 1 healthcare-10-02348-t001:** Intervention of Executive Functions with LEGO^®^-based therapy (also see Appendix A).

Session	Target	Activities	Material
Session 1 Free game with programming	Start of initial interaction and get the patient to familiarize with the material and achieve therapeutic interaction	- Colored blocks identification and free assembly. - Assembly and programming of robotics initial assembly. - In case of severe damage start with animal sets assembly. - Simple machine assembly. - Robotics challenge programming: forward and backward sequence at 10 s	- LEGO^®^ DUPLO^®^ bricks - WeDo 2.0 ^®^ Set - LEGO^®^ Simple machines Set - Bingo LEGO^®^ Education Bingo Set
Sessions 2 and 3 Working and visuospatial memory, inhibitory control	Gradually stimulate the CPDL and COM areas to generate changes in the selective attentional effort reversal process, inhibitory control and short-term memory	- Start with working memory exercise and 2 or 3 pieces blocks assembly. - Stimulate with a visuo-spatial working memory template, blocks of the same color - Spin assembly and challenge settings - Robot assembly and programming with challenge - Simple machine or robotic challenge disassembly and reassembly without support from a template or the therapist	- LEGO^®^ DUPLO^®^ bricks - WeDo 2.0 ^®^ Set - LEGO^®^ Simple machines Set - Assembled blocks template
Sessions 4 to 8 Working memory, inhibitory control, risk selection and planning	CPA, CPDL and COM areas stimulation to get changes in the effort investment process of the associated functions	- Start with working memory exercise and 3–4 pieces block assembly. - Stimulate with visuo-spatial working memory template, colored blocks. - Sort of color blocks by color and label. - Robot assembly and rogramming with challenge	- LEGO^®^ DUPLO^®^ bricks - WeDo 2.0 ^®^ Set - Assembled blocks template
Sessions 9 to 16 Follow-up and executive functions integration	Enablement of executive functions and monitoring of their development	- Memory assembly to 6 pieces. - Robot armed and assembly - Math adder for counting and regression - Complex challenge to solve with the robot - Concrete and abstract classification of animals set	- LEGO^®^ DUPLO^®^ bricks - WeDo 2.0 ^®^ Set or SPIKE^TM^ Set - LEGO^®^ Education Animals Set - LEGO^®^ Education More to Math Set

**Table 2 healthcare-10-02348-t002:** Patient characteristics.

Characteristic	Category	Control Group	Treatment Group
Sex ^1^	Male	6 (60%)	8 (57.14%)
	Female	4 (40%)	6 (42.86%)
Age ^2^		9.5 (8.0–12.25)	8.0 (6.25–10.0)
RACHS-1 ^1^	1	5 (50%)	1 (7.14%)
	2	3 (30%)	2 (14.29%)
	3	1 (10%)	5 (35.71%)
	4	1 (10%)	6 (42.86%)

^1^ Counts and percentages. ^2^ Median and interquartile range (IQR: 1st quartile–3rd quartile). Treatment = LEGO^®^-based therapy.

**Table 3 healthcare-10-02348-t003:** Scores and diagnostics of the neuropsychological executive functions battery in LEGO^®^-based therapy patients with congenital heart disease.

	Orbitomedial Cortex				Anterior Prefrontal Cortex				Dorsolateral Cortex				Executive Functions			
	Pretest		Posttest		Pretest		Posttest		Pretest		Posttest		Pretest		Posttest	
Pat	Score	Diag	Score	Diag	Score	Diag	Score	Diag	Score	Diag	Score	Diag	Score	Diag	Score	Diag
1	**43**	**SA**	**73**	**MMA**	**85**	**N**	**121**	**HN**	**74**	**MMA**	**98**	**N**	**51**	**SA**	**89**	**N**
2	**44**	**SA**	**91**	**N**	**70**	**MMA**	**109**	**N**	**108**	**N**	**124**	**HN**	**88**	**N**	**118**	**HN**
3	**92**	**N**	**136**	**HN**	117	HN	117	HN	123	HN	119	HN	122	HN	133	HN
4	**44**	**SA**	**86**	**N**	**107**	**N**	**117**	**HN**	77	**MMA**	**111**	**N**	**44**	**SA**	**108**	**N**
5	100	N	99	N	**48**	**SA**	**83**	**MMA**	70	**MMA**	**98**	**N**	**75**	**MMA**	**97**	**N**
6	102	N	85	N	114	N	96	N	86	N	101	N	92	N	97	N
7	**114**	**N**	**136**	**HN**	117	HN	110	N	97	N	106	N	**101**	**N**	**133**	**HN**
8	**125**	**HN**	**136**	**HN**	117	HN	117	HN	133	HN	134	HN	**72**	**MMA**	**133**	**HN**
9	**103**	**N**	**136**	**HN**	**96**	**N**	**117**	**HN**	108	N	89	N	**44**	**SA**	**133**	**HN**
10	**44**	**SA**	**136**	**HN**	**107**	**N**	**117**	**HN**	**66**	**SA**	**101**	**N**	**44**	**SA**	**133**	**HN**
11	92	N	103	N	117	HN	117	HN	99	N	92	N	100	N	95	N
12	**92**	**N**	**136**	**HN**	**110**	**N**	**117**	**HN**	116	HN	112	N	**114**	**N**	**127**	**HN**
13	**44**	**SA**	**115**	**N**	82	MMA	41	SA	67	SA	45	SA	47	SA	45	SA
14	43	SA	43	SA	63	SA	56	SA	45	SA	47	SA	45	SA	45	SA

SA = severe alteration (score of 69 or less); MMA = mild moderate alteration (score from 70 to 84); N = normal (score from 85 to 115); HN = high normal (score of 116 and greater). Pat = Patient; Diag = Diagnostic; (*n* = 14).

**Table 4 healthcare-10-02348-t004:** Scores and diagnostics of neuropsychological executive functions battery in control patients with congenital heart disease.

	Orbitomedial Cortex				Anterior Prefrontal Cortex				Dorsolateral Cortex				Executive Functions			
	Pretest		Posttest		Pretest		Posttest		Pretest		Posttest		Pretest		Posttest	
Pat	Score	Diag	Score	Diag	Score	Diag	Score	Diag	Score	Diag	Score	Diag	Score	Diag	Score	Diag
1	43	SA	43	SA	48	SA	41	SA	45	SA	45	SA	45	SA	45	SA
2	113	N	111	N	**63**	**SA**	**136**	**HN**	106	N	95	N	103	N	104	N
3	44	SA	44	SA	97	N	107	N	70	MMA	81	MMA	44	SA	44	SA
4	44	SA	44	SA	97	N	97	N	51	SA	69	SA	44	SA	44	SA
5	94	N	91	N	**50**	**SA**	**89**	**N**	86	N	111	N	**81**	**MMA**	**106**	**N**
6	124	HN	128	HN	**63**	**SA**	**94**	**N**	117	HN	100	N	117	HN	105	N
7	113	N	110	N	**74**	**MMA**	**95**	**N**	130	HN	113	N	130	HN	112	N
8	44	SA	44	SA	**112**	**N**	**122**	**HN**	75	MMA	77	MMA	44	SA	46	SA
9	114	N	103	N	103	N	103	N	79	MMA	82	MMA	83	MMA	84	MMA
10	120	HN	136	HN	103	N	110	N	**114**	**N**	**127**	**HN**	116	HN	133	HN

AS = severe alteration (score of 69 or less); MMA = mild moderate alteration (score from 70 to 84); N = normal (score from 85 to 115); HN = high normal (score of 116 and greater). Pat = Patient; Diag = Diagnostic; (*n* = 10).

## Data Availability

Not applicable.

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
