# Peer review of "Habilitation of Executive Functions in Pediatric Congenital Heart Disease Patients through LEGO^®^-Based Therapy: A Quasi-Experimental Study"

_healthcare, 2022, doi:10.3390/healthcare10122348_

Round 1

Reviewer 1 Report

Habilitation of executive functions in pediatric congenital heart disease patients through LEGO-based therapy

In the treatment group, in the second stage, the children were summoned once a week to carry out between seven and eighteen 60-minute interventions. The authors should mention why it is important more clearly.

Why the observed scores are higher as sessions progress

The authors should add the labels in x and y axis. Figure 1. Evaluation of execution of pediatric patients with congenital heart disease in LEGO®based therapy by sessions and RACHS-1 classification. Top: Observed profiles of each patient colored by his/her RACHS-1 level. Bottom: Mean profiles computed with the loess smoothing method (solid line) and 95% confidence intervals (shaded areas); colored lines represent the RACHS-1 level.

Why most patients who received LEGO-based therapy changed from alteration to normal category (with the exception of the anterior prefrontal cortex); for instance, in the orbitomedial cortex, in the control group, the 4 subjects categorized as alteration in the pretest continued in alteration in the posttest; however, for treatment group, 2 subjects categorized as alteration in the pretest continued in alteration, but 4 subjects categorized in alteration improved to normal category. What is the impact of LEGO therapy.

Do the therapy based on Lego® Education can only favor the group of neurons and processes associated with this cortex. Why not others?

The authors can refer

Multiobjectives for Optimal Geographic Routing in IoT Health Care System

InfusedHeart: A novel knowledge-infused learning framework for diagnosis of cardiovascular events

Author Response

  1. In the treatment group, in the second stage, children were summoned once a week for seven to eighteen 60-minute interventions. The authors should mention why this is important more clearly.

Answer: Thank you for your kind comments. The sessions were 60 minutes long since different functions were worked on in the same session, increasing the complexity of the exercises each week. The children took between seven and eighteen sessions depending on their ability to achieve the objectives, so the children with RACHS-1 levels 1 and 2 required fewer sessions, as they were able to assemble the more complex robots faster. This was explained in “2.4. Intervention Evaluation” section, lines 138-142.

  1. Why observed scores are higher as sessions progress.

Answer: The scores increased because the patients began with a low execution in the first sessions, and as they progressed, they obtained greater control of the material with greater motor capacity and in less execution time. Likewise, as the patients progressed in the sessions, they were able to solve more complex programming and robotics problems. It should be noted that the constancy of work in the sessions helps the neuronal group associated with executive functions to remain constantly stimulated, achieving greater concentration and irradiation, thus obtaining a long-term effect. This information was added to the discussion section in lines 342-346.

  1. Authors should add the labels on the x and y-axes. Evaluation of the execution of pediatric patients with congenital heart disease in LEGO®-based therapy by sessions and RACHS-1 classification. Top: Observed profiles of each patient colored by their RACHS-1 level. Bottom: Mean profiles calculated with the loess smoothing method (solid line) and 95% confidence intervals (shaded areas); colored lines represent the RACHS-1 level.

Answer: Clearer labels have been added to the x- and y-axes in Figure 1 and the top and bottom graphics.

  1. Why did the majority of patients who received LEGO-based therapy change from impaired to normal category (with the exception of the anterior prefrontal cortex); for example, in the orbitomedial cortex, in the control group, all 4 subjects categorized as impaired at pretest continued to be impaired at posttest; however, for the treatment group, 2 subjects categorized as impaired at pretest continued to be impaired, but 4 subjects categorized as impaired improved to the normal category. What is the impact of LEGO therapy.

Answer: In general, we observed improvement and a greater ability to solve problems in most of the patients in the experimental group. However, in the two cases in which there was no improvement, this could be because the patients finished the program, but sometimes, they missed some interventions, which led them to reschedule their session every 15 days, and with this, we observed that they could not maintain working memory or inhibitory control as the patients who were constant in all their sessions. As previously mentioned, the constancy of work in the LEGO sessions helps the neuronal group associated with executive functions to remain constantly stimulated, achieving greater concentration and irradiation, thus obtaining a long-term effect. This information was added to the discussion section in lines 346-352.

  1. Doing therapy based on Lego® Education can only favor the set of neurons and processes associated with this cortex. Why not others?

Answer: Indeed, Lego® Education sets can favor other cortexes and cognitive functions; however, in this approach, we evaluated executive functions, since the literature associates heart disease with oxygenation deficits in frontal and prefrontal arteries. Furthermore, our next objectives are to investigate other cortical regions and their associated cognitive functions in other pathologies.

The authors can refer:

  1. Multitargets for optimal geographic routing in the IoT healthcare system.

Answer: Thanks for your recommendation, in relation to the article Multiobjectives for optimal geographical routing in the IoT health care system, the proposed technology is very interesting for data management and routing of medical data; however, the technology used in LEGO® enablement uses interfaces through tablets and applications that connect via blothoot to a hub where it receives the instructions for the activation of the previously assembled robot.

  1. InfusedHeart: a new knowledge-based learning framework for the diagnosis of cardiovascular events.

Answer: With respect to this article, it is very important to develop technology that can be implemented in cardiovascular detection, and it would be interesting to implement it in future studies; however, in this court, we are studying patients who have already undergone heart surgery; in the future, we hope to study them prior to surgery.

Reviewer 2 Report

The present study presents quite interesting methodology and results. It is also quite robust, and well described. Nevertheless, the sample size (number of patients) was quite small. Although it could be difficult to increase, please address this as an important limitation of your study, by at least commenting on it in the discussion, conclusion and abstract sections. Also, please add a more thorough description of each used lego set, so that the reader can better understand the differences between sets, especially in what concerns complexity (if you could also show the images of the sets it would be even better).

Author Response

The present study presents quite interesting methodology and results. It is also quite robust and well described. However, the sample size (number of patients) was quite small. Although it might be difficult to increase, address this as an important limitation of your study, at least by commenting on it in the discussion, conclusion, and summary sections. Additionally, add a more detailed description of each lay set used, so that the reader can better understand the differences between the sets, especially in terms of complexity (if you could also show the images of the sets, it would be even better)

Answer:

A) Thank you for your comments; certainly, the sample was small since we had several inconveniences during the development of this work. Especially since it was carried out shortly before and during the COVID-19 pandemic, many patients who started therapy were unable to follow the process because their guardians or relatives lost their job or due to long distances from their homes to the institute, which limited their usual way of transportation, or due to the very restrictions of the situation. Indeed, it would be very difficult for us to increase the sample size of this study. However, we have scheduled a larger number of patients for future work and thus be able to obtain more robust conclusions about this therapy. We address some of this in the discussion (lines 358-362) and conclusion (line 370) sections. In the abstract section it was not possible due to the limited number of words allowed.

B) Thank you for your recommendation. Images for the material used were added to appendix 1 to better understand the reader.